# Effects of Responsiveness and Responsibility Parenting Factors on Family Mealtime Outcomes in Overweight African American Adolescents

**DOI:** 10.3390/nu16223874

**Published:** 2024-11-13

**Authors:** Haylee Loncar, Allison M. Sweeney, Taylor White, Mary Quattlebaum, Dawn K. Wilson

**Affiliations:** 1Department of Psychology, University of South Carolina, Columbia, SC 29201, USA; hloncar@sc.edu (H.L.); trw9@email.sc.edu (T.W.); mjq@email.sc.edu (M.Q.); 2Department of Biobehavioral Health and Nursing Science, College of Nursing, University of South Carolina, Columbia, SC 29201, USA; sweeneam@mailbox.sc.edu

**Keywords:** parenting, adolescents, family mealtime, African American, obesity

## Abstract

Background/Objectives: Family meals have been shown to be an important protective factor for positive health outcomes. This study assessed the associations of parenting factors with family mealtime among overweight African American adolescents over a period of 4 months. It was hypothesized that increases in warm and responsive parenting (parental responsiveness, parental responsibility) would be associated with increased frequency and quality of family mealtimes, while more demanding and controlling parenting (parental demandingness, parental monitoring) would be associated with a reduced frequency and quality of family mealtimes over time. Methods: Data from baseline to 16 weeks were collected from 241 African American adolescent–parent dyads (adolescent: *M_age_* = 12.8 ± 1.7 years; 64% female; *M_BMI%_
*= 96.6 ± 4.2) that participated in the Families Improving Together (FIT) for Weight Loss randomized controlled trial. Results: Multilevel models revealed significant positive main effects of parental responsiveness and parental responsibility (*p* < 0.05) on the increased frequency of family meals (*p* < 0.01). Significant two-way interactions also showed that parental responsiveness (*p* < 0.05) predicted improved quality of family mealtimes, whereas parental demandingness (*p* < 0.01) predicted reduced quality of family mealtimes from baseline to 16 weeks. Conclusions: Results from this study have important implications for African American adolescent obesity prevention and future family-based intervention program guidelines.

## 1. Introduction

Numerous studies have shown that family-related factors, including more frequent family meals, are associated with improved dietary intake and family functioning among adolescents [1,2,3,4]. Specifically, research suggests that more frequent family meals are associated with lower body mass index (BMI) and increased fruit and vegetable consumption among adolescents [1,5]. More frequent family meals have also been positively associated with family cohesion, support, and satisfaction [3,6]. Longitudinal studies indicate that family meals in adolescence also predict health outcomes in adulthood, including frequency of family mealtimes, fruit and vegetable consumption, and reduced risk of obesity [7,8]. However, African American families tend to have fewer family meals together compared to their non-minority counterparts [9,10]. African American adolescents also have a greater risk of living in obesogenic environments that may serve as a barrier to engaging in family meals [11]. Additionally, research has shown that overweight families report fewer positive interactions at mealtimes and greater time pressures around family mealtimes compared to normal-weight families [12,13]. Overall, few studies have focused on examining parenting factors that promote engagement in family mealtimes among overweight African American adolescents. Therefore, this study sought to address this gap by examining parenting factors associated with family mealtimes over time among African American families.

Family systems theory suggests that autonomy-supportive parenting practices are critical for promoting healthy outcomes among adolescents [14,15,16]. Specifically, parenting styles are comprised of parental attitudes and behaviors and are characterized by the degree of responsiveness (warmth) and demandingness (control) that parents practice with their adolescents [17,18]. In general, parenting styles have been conceptualized as authoritative (high responsiveness, high demandingness), authoritarian (low responsiveness, high demandingness), permissive (high responsiveness, low demandingness), and neglectful (low responsiveness, low demandingness) [17,18]. Authoritative parenting has been associated with a broad range of desirable health outcomes, including healthier weight-related outcomes, more nutritious dietary intake, and greater frequency of family mealtimes in adolescents [19,20,21,22,23,24,25,26,27]. Authoritative parenting has also been associated with increased family mealtime frequency among African American adolescents [20,25]; however, much of this work is cross-sectional in design. In contrast, authoritarian parenting has been associated with higher BMI in African American adolescents [28,29]. Though experts suggest that parenting style is generally a time-stable construct, insufficient data exists to conclude parenting style does not change over time [30]. Our group has previously published outcomes associated with BMI related to autonomy-supportive parenting in the Families Improving Together (FIT) for Weight Loss trial [31,32]. Thus, the current study fills the gap by expanding on the existing literature by providing a longitudinal perspective on the extent to which changes over time in parenting relate to family mealtime among overweight African American adolescents.

African American families are typically underrepresented in the research on associations between parenting practices and adolescent obesity-related outcomes. Some research suggests notable cultural variations in the parenting and feeding practices for adolescents [17,33,34]. Tamis-Lemonda et al., [34] highlighted variations in environmental factors and parents’ expectations during childhood that may contribute to cultural differences in parenting practices and youth outcomes. Specifically, some researchers have noted that authoritarian parenting can increase assertiveness and independence in adolescents, especially in low-income populations [34]. In addition, research has indicated that African American parents endorse high levels of authoritarian parenting practices and styles [35,36]. However, very little past research has examined associations of how parenting practices impact overall eating- and weight-related outcomes in an entirely African American adolescent sample over time.

Data analyzed within African American families show that authoritative, rather than authoritarian, parenting practices are associated with more desirable outcomes, such as greater academic achievement, fewer depressive symptoms, and fewer problem behaviors [37,38,39,40]. There is also increasing evidence that family-based interventions that address autonomy-supportive parenting (high responsiveness, low demandingness) are beneficial for African American adolescents who are overweight [14,15]. This distinction highlights the apparent gap in previous research that has not specifically studied African American populations or specifically overweight and obese youth.

Parent feeding practices have more recently been linked to family mealtime behaviors [14,20,25]. Parental feeding practices, which involve parents’ behaviors that affect their youth’s eating, are a prominent metric for understanding the relationship between parenting behaviors and family mealtime [20,25,41]. Parental responsibility for their adolescent’s diet highlights areas of support around diet, whereas parental monitoring of adolescent dietary intake involves aspects of control surrounding the adolescent’s diet. Unlike parenting style, prior research suggests that parenting practices, such as parental responsibility and monitoring of adolescent diet, may fluctuate over time [17]. However, the bulk of the existing literature is cross-sectional in design, limiting the interpretation of relationships and changes over time among parenting factors and adolescent diet [19,21,42]. Moreover, limited research exists that examines the association between parental feeding practices and family mealtime frequency or quality especially among African American populations. The cross-sectional literature provides support that feeding practices, responsibility, and monitoring are associated with adolescent weight-related outcomes [19,43,44], such that greater parental responsibility is related to healthier adolescent BMI, and greater parental monitoring is associated with higher adolescent BMI [43,44]. Additionally, research has suggested that African American parents tend to endorse high levels of parental monitoring, which appear to have mixed associations with the frequency of family mealtimes, adolescent BMI, and adolescent eating behaviors [19,23,25,36]. Given the limitations of cross-sectional evidence, the present study assessed these parenting feeding practices over a period of 4 months. In sum, familial influences from parenting style (responsiveness and demandingness) and feeding practices (responsibility and monitoring) may be linked to weight-related outcomes such as the frequency and quality of family mealtimes [14,15,20,25].

This proposed study examines the main effects of parenting factors (responsiveness, demandingness, responsibility, monitoring) in predicting adolescent family mealtime outcomes in overweight African American adolescents, over 4 months. To address the research aims, the present study presents secondary analyses from the Families Improving Together (FIT) for Weight Loss randomized controlled trial [31,32]. The FIT trial evaluated the efficacy of a family-based motivational weight-loss intervention as compared to a comprehensive health education program for African American families [31,32]. The primary outcome of the trial was adolescent zBMI, with parent BMI, moderate-to-vigorous PA (MVPA, adolescent and parent), light PA (LPA, adolescent and parent), and dietary outcomes (adolescent and parent) included as secondary outcomes. As reported previously, there were no significant intervention effects for BMI (parent or adolescent) or diet (parent or adolescent). There was a significant effect of the in-person motivational family weight loss program on parent LPA at 4 months [32]. Furthermore, complier average causal effects showed a significant effect at 4 months for parents on MVPA and a similar trend for adolescents [32]. Additionally, other secondary analyses from the FIT trial have revealed significant interactions between the in-person intervention and parenting style (responsiveness, demandingness) and parental feeding practices (restriction, concern about child’s weight) on the frequency of family mealtimes at 4 months [20]. The present study complements past findings from the FIT trial by focusing on the role of both family mealtime frequency and quality and evaluating the main effects of parenting style and parenting feeding practices across time while controlling for treatment condition and intervention effects. For the current study, it was hypothesized that increases in warm, responsive parenting (parental responsiveness, parental responsibility) would be associated with increased frequency and quality of family mealtimes, while more demanding, controlling parenting (demandingness, parental monitoring) would be associated with reductions in the frequency and quality of family mealtimes over time.

## 2. Materials and Methods

### 2.1. Participants

A total of 241 African American parent–adolescent dyads FIT randomized controlled trial [31,32] provided secondary data for this trial. Participants who took part in culturally relevant local events or festivals or who were part of the local pediatric clinics and parks and recreation partners [45] were recruited for the study. The advertisements emphasized that the program was a positive, family-fun health promotion program, not a weight loss program. Eligible families met the following criteria, which included having (1) an African American adolescent between 11–16 years of age, (2) an adolescent who was overweight or obese, as defined by having a BMI ≥ 85th percentile for their age and sex, (3) a caregiver willing to participate with the adolescent, and (4) access to the internet. Adolescents with medical or psychiatric conditions that would affect their diet or ability to exercise were excluded from the study. Caregivers and/or adolescents who were currently enrolled in another weight loss or health program were also excluded. All participants signed informed consent forms prior to participation and were given compensation for their participation in FIT.

### 2.2. Study Design

The FIT trial evaluated the efficacy of a family-based motivational weight-loss intervention as compared to a comprehensive health education program for African American families [31,32]. The current study includes longitudinal data collected at baseline, post-face-to-face intervention (2 months), and post-online intervention (4 months). The FIT trial used a 2 × 2 factorial design, in which participants were initially randomized to a 2-month, in-person motivational family weight loss program or a comprehensive health program, and then rerandomized to either a 2-month online tailored intervention or to an online control condition. While the FIT trial was an intervention, this study controls for treatment condition and intervention effects. Full methods and the primary outcomes of Project FIT, including missing data, have been published elsewhere [31,32]. Overall, the retention rate for the FIT trial was 72%. Project FIT was approved by the University of South Carolina Institutional Review Board.

### 2.3. Procedures

Anthropometric measurements (height and weight) and psychosocial surveys were obtained from parents and adolescents. A Seca 880 digital scale (Seca Ltd., Birmingham, UK) was used to measure weight, and a Shorr height board (Weight and Measure, LLC., Olney, MD, USA) was used to measure height. Adolescent BMI was calculated using these measures with the Center for Disease Control (CDC) growth charts [46], then standardized to BMI z-scores (zBMI) using the statistical analysis system (SAS) program (version 9.4TS1M8). All anthropometric, psychosocial measures and family mealtime measures were completed at baseline, post-group timepoint (8 weeks), and post-online timepoint (16 weeks). Participants were incentivized for their time at the conclusion of each timepoint for a total of $110. Project FIT followed appropriate informed consent procedures.

### 2.4. Measures

#### 2.4.1. Demographic Information

Demographic information was obtained through self-reported by parent or adolescent psychosocial surveys. These measures included parent and adolescent age, parent and adolescent sex, parent annual income, parent education, parent marital status, and number of children in the household.

#### 2.4.2. Adolescent-Reported Parenting Style

Parenting style was measured from an adolescent self-report measure, the Authoritative Parenting Index (API), using six items [47]. The API consisted of two subscales of responsiveness and demandingness. Responses were reported using a 5-point Likert scale ranging from “not at all like them” to “exactly like them”. A sample item included “My parents make me feel better when I am upset”. The demandingness and responsiveness subscales have been shown to be reliable for adolescents (α = 0.77 and 0.85, respectively). Previous studies have demonstrated the construct validity of this measure [47,48].

#### 2.4.3. Parent-Reported Child Feeding Questionnaire

The Child Feeding Questionnaire (CFQ) [41] was used to assess parental feeding practices and feeding styles. The parent-report scale consisted of five subscales measuring five dimensions of feeding: parental responsibility, restriction, concern, monitoring, and pressure to eat. Parental responsibility and parental monitoring were assessed in this study. This scale has been validated for use with adolescents, and each dimension is sufficiently reliable [49]. Items in this questionnaire have been modified to reflect the adolescent’s perspective on their parent’s feeding practices. Responses for each dimension were scored on a 5-point Likert scale.

Adolescent-Reported Parental Responsibility

The CFQ responsibility dimension consisted of three items and assessed parental feeding responsibility from the adolescent’s perspective. Sample questions included “When home, how often is my parent responsible for preparing my meals?” and “How often is my parent responsible for deciding if I have eaten the right kind of foods?”. Responses ranged from “1 = never” to “5 = always”.

b.Adolescent-Reported Parental Monitoring

The CFQ monitoring dimension consisted of three items that evaluate parental monitoring of adolescent diet from the adolescent’s perspective. Sample questions included “How often does my parent keep track of the sweets (candy, ice cream, pies, pastries) that I eat?” and “How often does my parent keep track of the high-fat foods that I eat?”. Responses ranged from “1 = never” to “5 = always”.

#### 2.4.4. Frequency and Quality of Family Mealtimes

For the frequency of family mealtime measure, adolescents reported the number of meals they had with their family during a typical week using a validated scale [50]. Response choices ranged from 1 to 6: 1 (never), 2 (1–2 times), 3 (3–4 times), 4 (5–6 times), 5 (7 times), and 6 (more than 7 times). The scale has been used to assess mealtime outcomes in diverse racial populations and has shown construct validity [50]. For the quality of family mealtime measure, adolescents rated their family meal environment by indicating how strongly they agreed with statements regarding the positive climate and conversations during family meals. The scale was validated with a diverse population of youth and was reliable (α = 0.73) [50]. There were four items on the measure, which included statements such as “I enjoy eating meals with my family.” and “In my family, dinner time is about more than just getting food, we all talk to each other”. Item responses ranged from 1 to 4: 1 (strongly disagree), 2 (somewhat disagree), 3 (somewhat agree), and 4 (strongly agree).

### 2.5. Data Analytic Plan

#### Multilevel Model Building

A growth curve analysis approach was used to allow for the estimation of effects occurring at multiple time points within an individual. Models were developed with the R statistical software package, version 4.2.2 (The R Foundation for Statistical Computing, Vienna, Austria), using a stepped approach. The model-building procedure involved testing a series of models with increasing complexity to predict adolescent family mealtime and to account for the nesting of participants within treatment groups. To determine which model best fits the data, a series of chi-square difference tests were conducted with family mealtime frequency as the outcome. If the more complex model did not yield a significantly better fit, then the simpler model was retained. This approach indicated that the best-fitting model included a fixed effect for time and a random effect for groups. This modeling approach was used for all subsequent analyses and is consistent with the approach used in prior FIT trial analyses [32].

The best-fitting model (i.e., the model with the fixed effect for time and random effect for group) was incorporated for a series of model comparisons that included theory-based predictors. Using a hierarchical approach, a series of chi-square difference tests compared a covariate-only model, the addition of main effects, and the addition of the two-way interaction terms between parenting factors and time. Covariates included adolescent age, adolescent sex, parent education, parent BMI, and a main effect for time (0 = baseline, 1 = 8 weeks, 2 = 16 weeks). These covariates have been the standard used in previous analyses for the FIT trial [20,32]. Predictor variables included parenting variables (parental responsiveness, demandingness, responsibility for adolescent diet, and monitoring of adolescent diet). All predictor variables were z-scored. Two-way interactions between the predictors and time were used to test the study hypothesis.

All multilevel model assumptions and case diagnostics were tested before running outcome analyses. To address the assumption of normality, histograms of the standardized residuals were assessed, and data were found to be normally distributed. Scatterplots of the standardized residuals and predicted values were evaluated, and independent variables exhibited homoscedasticity. Additionally, scatterplots were used to examine variability between groups and confirmed that error was randomly distributed across levels of each model predictor. A Durbin–Watson test was used to assess the independence of errors. A Cook’s distance measure was used to check for influential points in the data, and no cases were deemed to be significantly influential, so the final models included all 241 participants. Bivariate correlations between independent variables were used to assess potential multicollinearity.

Adolescent family mealtime was measured over time (baseline to 4 months), and longitudinal assumptions, including stability, stationarity, and equilibrium, were tested. The stability of the mean over time was examined by comparing means of family mealtime at both time points. Stationarity, which assumes that family mealtime measurements were obtained in the same manner at baseline and post-intervention, has been met due to the strict protocol for obtaining measurements by trained staff during the intervention. Equilibrium, which assumes temporal stability in the patterns of covariance and variance among variables, was tested by comparing variance and covariance scores across the two measurements of family mealtime. These preliminary analyses demonstrated that multilevel modeling assumptions were met.

Missing data in the FIT trial were assumed to be missing at random. Multiple imputations were used to address missing data using the MICE package in R. All outcomes, demographic data, and variables of theoretical importance, including the key variables assessed in the present analyses, were included in the imputation to minimize the likelihood of biased estimates and meet missing at random assumptions. A total of 20 datasets were imputed, and one random imputation was selected for the analyses of the proposed study [20].

## 3. Results

### 3.1. Demographics

The demographics of the study sample are presented in Table 1. Additionally, Table 2 presents the means and standard deviations for all predictor and outcome variables across all time points.

### 3.2. Correlation Analyses

Results of the correlational analyses indicated that several covariate, predictor, and outcome variables were correlated across the three time points. Among baseline variables, the strongest correlations included those between parental demandingness and parental responsiveness (*r* = 0.48) and between parental monitoring and parental responsibility (*r* = 0.53). Among post-group intervention variables (8 weeks), the strongest positive correlations were those between parental demandingness and parental responsiveness (*r* = 0.46) and between parental responsiveness and the frequency of family meals (*r* = 0.44). The strongest correlations among post-online variables (16 weeks) included those between parental responsiveness and parental demandingness (*r* = 0.42) and between parental responsiveness and the frequency of family meals *(r* = 0.40). Across time points, the strongest correlations included those between baseline and post-group intervention responsiveness (*r* = 0.52) and between post-group intervention and post-online demandingness (*r* = 0.44).

### 3.3. Frequency of Family Mealtimes

Using a hierarchical approach, a series of chi-square difference tests was conducted to compare a covariate-only model (Model 1), the addition of main effects (Model 2), and the addition of the two-way interaction terms between parenting factors and time (Model 3). For the frequency of family mealtime outcomes, the best-fitting model was Model 2 (main effects only; *ꭓ*^2^ (14) = 181.23, *p* < 0.01). In other words, model fit was not improved with the addition of two-way interactions (*ꭓ*^2^ (18) = 6.69, *p* = 0.22). However, individual two-way interaction terms were included in the final model to test the hypotheses, and all significant effects were interpreted.

Table 3 (below) shows a significant main effect of parental responsiveness, *Estimate* = 0.41, *SE* = 0.10, *p* < 0.01, which was not moderated by time, indicating that greater parental responsiveness was associated with greater frequency of family mealtimes, and this association remained stable across time. Furthermore, there was a main effect of parent responsibility, *Estimate* = 0.20, *SE* = 0.10, *p* = 0.045, which was not moderated by time, indicating that greater parental responsibility was associated with greater frequency of family mealtimes, and this association remained stable across time. There were no significant interactions or other main effects.

### 3.4. Quality of Family Mealtimes

For the quality of family mealtime outcome, the best-fitting model was Model 3 (two-way interactions with time; *ꭓ*^2^ (18) = 16.78, *p* < 0.01).

Results for the quality of family mealtimes indicated a significant two-way interaction between parental responsiveness and time, *Estimate* = 0.17, *SE* = 0.08, *p* = 0.026 (Table 4). Among those with low parent responsiveness, the quality of family meals decreased over time, whereas among those with high parent responsiveness, the quality of family meals increased over time (Figure 1). There was also a significant two-way interaction between parental demandingness and time, *Estimate* = −0.25, *SE* = 0.08, *p* < 0.01. Among those with high parent demandingness, the quality of family meals decreased over time, whereas among those with low parent demandingness, the quality of family meals increased over time (Figure 2). Furthermore, there was a significant main effect for parental responsibility, *Estimate* = 0.33, *SE* = 0.17, *p* = 0.05, such that greater parental responsibility was associated with greater quality of family mealtimes, and this association remained stable across time.

## 4. Discussion

This study examined the relationships between parenting factors and adolescent family mealtime outcomes in African American adolescents with overweight and obesity over a period of 4 months. There were significant main effects of responsiveness and responsibility on the frequency of family mealtimes, indicating that greater parental responsiveness and greater responsibility were associated with more frequent family meals, with these effects being relatively stable over time. Furthermore, both responsiveness and demandingness were moderated by time when predicting the quality of family mealtimes, such that lower parental responsiveness and higher parental demandingness were associated with reduced quality of family mealtimes over time. Overall, these findings are consistent with the prediction that increases in warm and responsive parenting (parental responsiveness, parental responsibility) would be associated with increased frequency and quality of family mealtimes, while more demanding and controlling parenting (demandingness, parental monitoring) would be associated with reductions in the frequency and quality of family mealtimes over time.

While authoritative parenting, including responsiveness and responsibility, has been associated with a broad range of desirable health outcomes, including healthier weight-related outcomes, more nutritious dietary intake, and greater frequency of family mealtimes in adolescents [19,20,21,22,23,24,25], past research has primarily been cross-sectional and has not focused specifically on African American adolescents with overweight or obesity. For instance, Berge and colleagues [51] found that when children and adolescents enjoyed family mealtimes, they were less likely to be overweight. In fact, a recent meta-analysis concluded that the quality, even more so than the quantity, of family mealtimes is related to adolescent health [2]. This aligns with established findings that positive familial interactions, such as those during family meals, promote child and adolescent health [14]. Parents who foster a warm and encouraging parenting style may facilitate more support for their adolescents at family meals and use family meals as a means to connect with their adolescents through positive family communication and bonding. Conversely, a higher quality of family mealtimes may translate to increased perceptions of parental responsiveness through positive family factors at mealtimes, such as atmosphere, communication, and family cohesion [3,6,52,53]. The present study adds to past research by demonstrating that the positive effects of responsiveness and responsibility on the frequency of family mealtimes are relatively stable, while the effects of responsiveness and demandingness on family mealtime quality increased over time. These findings suggest that parenting factors may have differential effects on family mealtime frequency versus quality and that future interventions may benefit from targeting different aspects of parenting depending on the outcome of interest.

Taken together, the findings on parental responsiveness are particularly meaningful when considering the ripple effects of family mealtime. Specifically, previous studies have demonstrated that eating meals together as a family is a predictor of adolescent health outcomes [4,5,25,54]. For instance, in their recent meta-analysis, Robson and colleagues [4] described evidence that eating together as a family more often is associated with positive dietary outcomes such as increased fruit and vegetable intake. Other studies have shown direct relationships between the frequency of family mealtimes and other health outcomes, such as adolescent BMI [5]. Furthermore, Berge and colleagues [55] found that a greater frequency of family mealtimes in adolescence went on to predict health outcomes 10 years later, demonstrating significantly better health for individuals who had more meals with their family during youth. The findings of the current study provide support for the positive effects of warm and responsive parenting (high parental responsiveness) and the potentially negative effects of demanding and controlling parenting (high demandingness), which may have ripple effects in improving additional adolescent health-related outcomes.

This study also found that adolescents with more demanding parents perceived mealtime to be lower quality over time, while adolescents with less demanding parents reported higher quality mealtime over time. This is consistent with the previous literature on the impact of demandingness on familial interactions [2,5,15]. Previous research has demonstrated that parents with overweight adolescents tend to exercise more control over their adolescents’ diet and eating habits at family meals compared to parents with normal-weight adolescents [13,56]. Additionally, parents’ attempts to gain control over their adolescents’ diet or weight by endorsing more controlled parenting practices may impede the development of autonomous dietary and health habits and limit opportunities for adolescents to practice independence in eating healthily at mealtimes [57,58,59]. This result is also meaningful as it provides further evidence for the importance of addressing parental demandingness in future family-based interventions as an impediment to positive quality family mealtimes over time, especially in African American families, which has been understudied.

### 4.1. Strengths and Limitations

There are a few limitations of this study that should be noted. Regarding design, future research may expand on the current findings by observing relationships across longer timespans or using causal study designs, which may provide additional insights regarding the temporal stability of variables, moderation effects, mediational effects, and direct relationships between parenting factors and adolescent health outcomes. However, even given this limitation, this study provides important information that informs future guidelines for parenting interventions in African American adolescents with overweight or obesity.

A significant strength of this study includes the focus on African American adolescents with overweight or obesity between the ages of 11 and 16 and a parent or caregiver, which is an underrepresented population. Very few studies exist that adequately represent African American families, and few studies examine relationships in entirely overweight or obese samples. While the present study was implemented in the US Southeast, it will be important for future studies to include other geographic regions and consider the heterogeneity within African American communities. Lastly, this study was over 4 months and allowed for the exploration of temporal stability and change in parenting factors in predicting adolescent mealtime outcomes. Findings demonstrate that parenting factors (e.g., enhancing responsiveness and reducing demandingness) should be considered as potential intervention targets in future interventions aiming to improve family mealtime, nutrition, and other weight-related outcomes among African American families.

### 4.2. Future Directions

An interesting aspect of the present study is the incorporation of family mealtime outcomes. Factors of adolescent health, the frequency and quality of family mealtimes, are proving to be significant predictors of child and adolescent health outcomes [2,5,55]. An expansion of the current study may consider the direct relationships between family mealtime variables and other adolescent health outcomes. Specifically, findings in the present study may guide future prevention research efforts and guidelines for obesity prevention and intervention programs in minority youth by focusing on the differential role of parenting factors. Given the associations of parenting factors related to responsiveness and responsibility on family mealtime outcomes, this could inform future guidelines for interventions in minority youth. Strong evidence exists demonstrating that lifestyle interventions involving children with obesity comorbidities in ethnic minority populations, such as physical activities and dietary interventions, are successful in reversing obesity as well as cardiometabolic disease risks [60,61]. Furthermore, interventions combining cultural contextual factors and engagement with families have also been shown to be effective in high-risk pediatric minority populations. Complementing these lifestyle programs with positive parenting skills may be a promising approach for future obesity prevention and intervention efforts. Specifically, this study suggests that facilitating more responsiveness and responsibility among parents in future interventions may be an effective approach for increasing the frequency of family mealtimes among underserved families. Future intervention guideline efforts should work to sustain preventative health impact, such as integrating parenting mealtime practices through the involvement of the community, families, and stakeholders, including healthcare professionals, nutritionists, exercise science specialists, and policymakers [60].

## 5. Conclusions

In summary, childhood overweight and obesity continue to be a significant health concern and especially affect African American families [62]. Identifying parental factors, such as responsiveness and responsibility, that influence adolescent weight and family mealtime is essential in creating efficacious interventions to address overweight and obesity in these understudied populations. The current study filled a gap in the literature by assessing the associations between parenting factors and adolescent mealtime outcomes specifically among overweight African American adolescents over time. A novel and important finding was the significant positive effects of parental responsiveness (nurturance) and responsibility (supportive environmental influences) on the quality of family mealtimes over time. A growing body of literature supports the importance of family mealtime and its implications for youth development. Continued research is needed to further investigate the family environment and its relationships with African American adolescents’ health and development.

## Figures and Tables

**Figure 1 nutrients-16-03874-f001:**
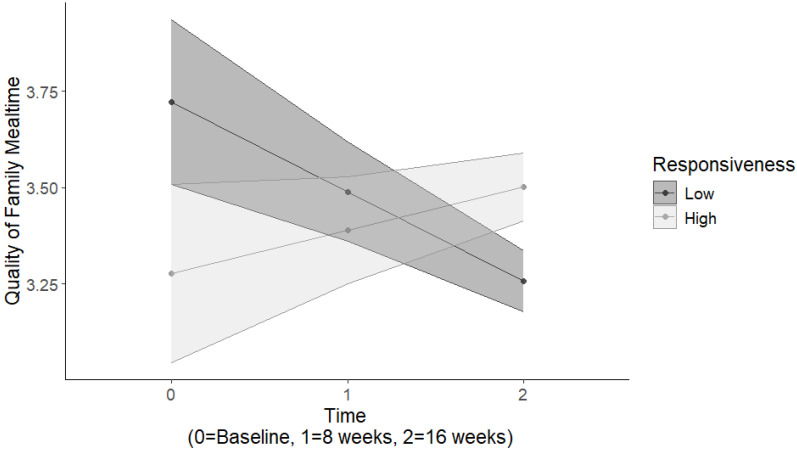
Parental responsiveness-by-time interactions predicting the quality of family mealtimes.

**Figure 2 nutrients-16-03874-f002:**
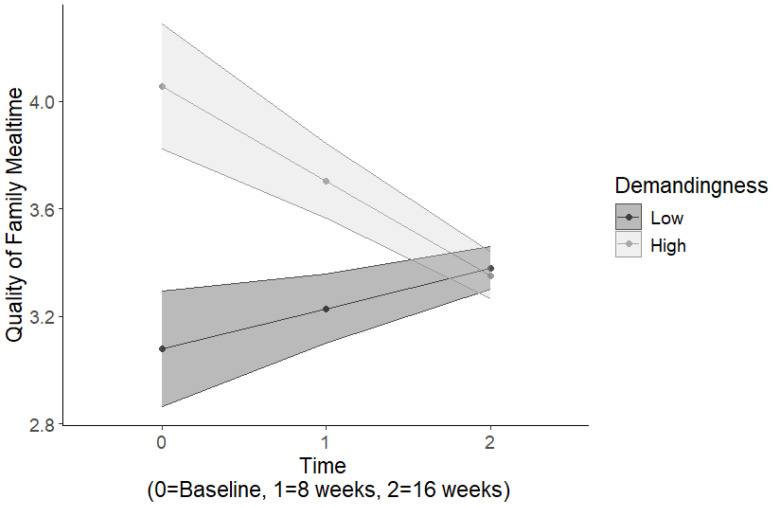
Parental demandingness-by-time interactions predicting the quality of family mealtimes.

**Table 1 nutrients-16-03874-t001:** Descriptive baseline data of study sample (*n* = 241).

Variable	Value
Adolescent Age *M*(*SD*)	12.83 (1.75)
Adolescent Sex (Female), *N*%	153 (64%)
Parent Education, *N*%	
9 to 11 Years	6 (2.5%)
12 Years	33 (13.7%)
Some College	99 (41.1%)
4 Year College	45 (18.7%)
Professional	55 (22.8%)
Parent BMI *M*(*SD*)	37.49 (8.34)
Parent Age *M*(*SD*)	43.18 (8.65)
Parent Income, *N*%	
Less than $10,000	36 (14.9%)
$10,000–$24,000	46 (19.1%)
$25,000–$39,000	65 (27.0%)
$40,000–$54,000	31 (12.9%)
$55,000–$69,000	21 (8.7%)
$70,000–$84,000	12 (5.0%)
$85,000 or greater	24 (10.0%)
Parents Married, *N*(%)	83 (34.4%)
Children in Home, *M*(*SD*)	2.05 (1.20)
Adolescent BMI Percentile *M*(*SD*)	96.61 (4.25)

**Table 2 nutrients-16-03874-t002:** Predictor means and standard deviations across time points, *M*(*SD*).

Variable	Baseline (0 Weeks)	Post-Intervention (8 Weeks)	Post-Online (16 Weeks)
Responsiveness	4.38 (1.02)	4.32 (1.26)	4.19 (1.38)
Demandingness	5.20 (0.80)	5.02 (0.86)	4.91 (1.12)
Responsibility	2.85 (0.51)	2.86 (0.53)	2.99 (0.52)
Monitoring	2.82 (0.98)	3.06 (0.97)	3.18 (0.97)
Freq. Family Meals (meals/week)	3.46 (1.61)	3.44 (1.55)	3.24 (1.52)
Qual. Family Meals	3.26 (0.65)	3.35 (0.61)	3.17 (0.76)

Note. “Freq. Family Meals” represents the frequency of family meals, and “Qual. Family Meals” represents the quality of family meals. Scores range from 1 to 5 for all parenting factors with higher values representing more endorsement. Scores range from 1 to 4 for quality of family mealtime with higher scores representing greater quality.

**Table 3 nutrients-16-03874-t003:** Multilevel model predicting the frequency of family mealtimes.

	Estimate	SE	*p*-Value	Lower 95% CI	Upper 95% CI
Intercept	−0.18	0.11	0.109	−0.40	0.04
Group Randomization	0.01	0.08	0.871	−0.14	0.16
Online Randomization	0.06	0.07	0.380	−0.07	0.19
Child Age	−0.01	0.02	0.452	−0.05	0.02
Child Sex	0.11	0.07	0.122	−0.03	0.25
Parent Education	0.27	0.07	<0.01 *	0.13	0.40
Parent BMI	0.00	0.00	0.424	0.00	0.01
Time	−0.02	0.04	0.703	−0.09	0.06
Responsiveness	0.41	0.10	<0.001 *	0.21	0.60
Demandingness	0.00	0.10	0.966	−0.20	0.19
Responsibility	0.20	0.10	0.045 *	0.01	0.40
Monitoring	−0.02	0.10	0.839	−0.22	0.17
Time × Responsiveness	−0.05	0.05	0.289	−0.14	0.04
Time × Demandingness	0.05	0.04	0.299	−0.04	0.13
Time × Responsibility	−0.06	0.04	0.192	−0.15	0.03
Time × Monitoring	0.08	0.04	0.062	0.00	0.17

Note: The notation * indicates a significant *p*-value.

**Table 4 nutrients-16-03874-t004:** Multilevel model predicting the quality of family mealtimes.

	Estimate	SE	*p*-Value	Lower 95% CI	Upper 95% CI
Intercept	3.32	0.19	<0.001	2.96	3.68
Group Randomization	0.21	0.11	0.065	−0.01	0.44
Online Randomization	0.11	0.11	0.350	−0.11	0.32
Child Age	−0.12	0.03	<0.001 *	−0.19	−0.06
Child Sex	0.08	0.12	0.505	−0.15	0.31
Parent Education	0.17	0.12	0.150	−0.06	0.39
Parent BMI	0.01	0.01	0.095	0.00	0.02
Time	−0.09	0.07	0.185	−0.23	0.04
Responsiveness	−0.22	0.17	0.191	−0.55	0.11
Demandingness	0.49	0.17	0.004 *	0.16	0.82
Responsibility	0.33	0.17	0.054 *	0.00	0.67
Monitoring	0.21	0.17	0.218	−0.12	0.54
Time × Responsiveness	0.17	0.08	0.026 *	0.02	0.32
Time × Demandingness	−0.25	0.08	<0.001 *	−0.40	−0.10
Time × Responsiblity	−0.10	0.08	0.211	−0.24	0.05
Time × Monitoring	−0.05	0.08	0.530	−0.19	0.10

Note: The notation * indicates a significant *p*-value.

## Data Availability

The dataset used during the current study is available from the corresponding author upon reasonable request.

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
