# Peer review of "Effects of Responsiveness and Responsibility Parenting Factors on Family Mealtime Outcomes in Overweight African American Adolescents"

_nutrients, 2024, doi:10.3390/nu16223874_

Round 1
Reviewer 1 Report
Comments and Suggestions for Authors
Introduction:
· Not clear what behavior self-efficacy is being tied to on lines 65-76. Is it self-efficacy to consume family meals together, self-efficacy to eat fruits and vegetables etc? Need to further define the specific action the self-efficacy is being tied to and ensure the literature being reviewed matches the construct being studied.
· Good use of theory to frame research purpose and questions.
· There is an abundance of literature on the effects of parenting styles on eating outcomes in adolescents in addition to pre-adolescents. It is not accurate to say that most of the literature is in pre-adolescents. Also, your sample includes both adolescents and pre-adolescents.
· There are also some reported differences in parenting styles and perceptions of parenting styles and their effects on diet-related behaviors based on racial/ethnic backgrounds. Since you are specifically looking at African American families this literature should be included. The lack of cultural context for the paper is a severe limitation.
· Line 120 – again it is not clear what you mean by self-efficacy because it hasn’t been connected to a specific action
Methods:
· 241 dyads were enrolled in the program – how many completed the program? How much data was missing? This information is important to understanding the validity of the data being presented.
· The self-efficacy for health behavior scale information is confusing – the self efficacy for exercise scale was used which has nothing to do with diet or family meals. Based on the definition of self-efficacy using self-efficacy for exercise is not conceptually appropriate. In addition a correlation of between 0.57-.73 between self-efficacy for exercise and nutrition behaviors is too low to use as a replacement for a scale on dietary self-efficacy.
Results and Discussion:
· Would like to see further discussion of relevant covariates and their impact on the analyses
· There are serious conceptual limitations to this study that are not discussed.
Author Response
Reviewer #1
- Introduction: Not clear what behavior self-efficacy is being tied to on lines 65-76. Is it self-efficacy to consume family meals together, self-efficacy to eat fruits and vegetables etc? Need to further define the specific action the self-efficacy is being tied to and ensure the literature being reviewed matches the construct being studied.
Response: Thank you for raising this issue. Based on this comment and other related comments below, we have removed self-efficacy as a variable from the study along with conceptual discussions of self-efficacy from the manuscript.
- Good use of theory to frame research purpose and questions.
Response: We are very appreciative of this comment.
- There is an abundance of literature on the effects of parenting styles on eating outcomes in adolescents in addition to pre-adolescents. It is not accurate to say that most of the literature is in pre-adolescents. Also, your sample includes both adolescents and pre-adolescents.
Response: Thank you for this comment. Our sample ranges from ages to 11 to 16 years old with a mean age of 12.83 and standard deviation of 1.75, which does include preadolescent to adolescent youth. We have revised the Introduction to better represent the current state of the literature as suggested (see revised introduction, paragraphs 2-5).
- There are also some reported differences in parenting styles and perceptions of parenting styles and their effects on diet-related behaviors based on racial/ethnic backgrounds. Since you are specifically looking at African American families this literature should be included. The lack of cultural context for the paper is a severe limitation.
Response: As suggested we revised the introduction to specifically highlight studies on parenting styles and health-related outcomes among African American youth and the importance of considering cultural context (see revised introduction, paragraphs 2-4).
- Line 120 – again it is not clear what you mean by self-efficacy because it hasn’t been connected to a specific action
Response: As mentioned above, we have revised the manuscript to remove the reference to self-efficacy from the entire paper.
- Methods: 241 dyads were enrolled in the program – how many completed the program? How much data was missing? This information is important to understanding the validity of the data being presented.
Response: Thank you for this comment. We note that all the data from 241 dyads were used in this study, as missing data was imputed (See last paragraph under Data Analytic Plan – Multi-Level Model Building). We also now include the retention rate of the study and refer the reading to the outcome study for full details on missing data (See study design paragraph under Materials and Methods).
- The self-efficacy for health behavior scale information is confusing – the self efficacy for exercise scale was used which has nothing to do with diet or family meals. Based on the definition of self-efficacy using self-efficacy for exercise is not conceptually appropriate. In addition a correlation of between 0.57-.73 between self-efficacy for exercise and nutrition behaviors is too low to use as a replacement for a scale on dietary self-efficacy.
Response: As noted already have removed the self-efficacy from the entire manuscript.
- Results and Discussion: Would like to see further discussion of relevant covariates and their impact on the analyses
Response: Thank you for this feedback. The decision to use covariates was guided by prior research (i.e., known factors that influence dietary and mealtime outcomes), and the need to be consistent with our previous analyses of the larger FIT trial (See paragraph 2 under Data Analytic Plan- Multi-Level Model Building).
- There are serious conceptual limitations to this study that are not discussed.
Response: After removing self-efficacy, we feel that the limitations have been sufficiently addressed (See paragraph 6 of the Discussion).
Reviewer 2 Report
Comments and Suggestions for Authors
This study is very interesting, but there are issues that need to be reconsidered and corrected.
1. The introduction is too long, please summarize it briefly.
2. Although this study was about parent-child interactions and diet, the authors should present and discuss data regarding changes in parent and child BMI during the observation period. The authors should demonstrate the impact of changes in household eating habits on obesity and overweight.
3. The authors should refer to reports with data from non-obese households and discuss differences in eating habits between non-obese and obese households.
4. The authors should provide data on whether the study subjects were aware that they were obese and at high risk for metabolic diseases in the future. If there is no data, they should cite the paper and discuss it.
Author Response
Reviewer #2
This study is very interesting, but there are issues that need to be reconsidered and corrected.
Response: We greatly appreciate your feedback and have responded to the issues below.
- The introduction is too long, please summarize it briefly.
Response: We have refocused the introduction to more clearly include the cultural context and importance of examining longitudinal relationships between parenting factors and dietary outcomes related to weight, dietary intake and family mealtime (see paragraphs 2-5). This revised introduction is more concise given that the background studies on self-efficacy have not been removed from the paper.
- Although this study was about parent-child interactions and diet, the authors should present and discuss data regarding changes in parent and child BMI during the observation period. The authors should demonstrate the impact of changes in household eating habits on obesity and overweight.
Response: Thank you for this comment. We note that the outcome paper has been previously published examining BMI outcomes and the need to address the cultural context as well as longitudinal relationships between parenting factors and adolescent weight-related outcomes including family mealtime in overweight African American adolescents (See Introduction paragraphs 2-6 and Study Design paragraph under Materials and Methods).
- The authors should refer to reports with data from non-obese households and discuss differences in eating habits between non-obese and obese households.
Response: Thank you for bringing our attention to this data. We have added in some general discussion that includes normal weight populations in addition to the focus on overweight populations as suggested (see paragraph 1 of the introduction and paragraph 4 of Discussion).
- The authors should provide data on whether the study subjects were aware that they were obese and at high risk for metabolic diseases in the future. If there is no data, they should cite the paper and discuss it.
Response: We have noted in that our recruitment efforts focused on promoting positive family-fun and health promotion to reduce the stigma of being overweight for youth and families (see Participants paragraph under Materials and Methods).
Round 2
Reviewer 2 Report
Comments and Suggestions for Authors
In response to my comment that additional data should be added, the authors replied that they had already reported the data in their paper.
When authors report that they have already reported data in their own paper, they should cite the data in the text in a way that makes it clear that they have already published the data, e.g., "We have already reported that..."
Please note the following: If the data I asked the authors to present was from another group and not their own data, then the paper would be judged as data-deficient.
Author Response
Thank you for your comments about more clearly cited our past studies which have already extensively look at BMI as an outcome in the FIT Trial. We now cite this information on lines 62-64 in the introduction to make it more clear how this paper expands on that previous work and fills a gap in the literature. We hope this is satisfactory.
Sincerely,
Dawn